

# E-MFNN: an emotion-multimodal fusion neural network framework for emotion recognition

Zhuen Guo[1], Mingqing Yang[1], Li Lin[1], Jisong Li[1], Shuyue Zhang[2,3], Qianbo He[2,3], Jiaqi Gao[2,3], Heling Meng[1], Xinran Chen[1], Yuehao Tao[1] and Chen Yang[1]

[1] School of Mechanical Engineering, Guizhou University, Guiyang, Guizhou, China
[2] University of North Alabama, Florence, AL, United States
[3] North Alabama International College of Engineering and Technology, Guizhou University, Guiyang, Guizhou, China

Corresponding authors
Mingqing Yang,
yangmq99@163.com
Li Lin, 715842042@qq.com

## ABSTRACT

Emotional recognition is a pivotal research domain in computer and cognitive science. Recent advancements have led to various emotion recognition methods, leveraging data from diverse sources like speech, facial expressions, electroencephalogram (EEG), electrocardiogram, and eye tracking (ET). This article introduces a novel emotion recognition framework, primarily targeting the analysis of users' psychological reactions and stimuli. It is important to note that the stimuli eliciting emotional responses are as critical as the responses themselves. Hence, our approach synergizes stimulus data with physical and physiological signals, pioneering a multimodal method for emotional cognition. Our proposed framework unites stimulus source data with physiological signals, aiming to enhance the accuracy and robustness of emotion recognition through data integration. We initiated an emotional cognition experiment to gather EEG and ET data alongside recording emotional responses. Building on this, we developed the Emotion-Multimodal Fusion Neural Network (E-MFNN), optimized for multimodal data fusion to process both stimulus and physiological data. We conducted extensive comparisons between our framework's outcomes and those from existing models, also assessing various algorithmic approaches within our framework. This comparison underscores our framework's efficacy in multimodal emotion recognition. The source code is publicly available at https://figshare.com/s/8833d837871c78542b29.

## INTRODUCTION

Emotional recognition simulates and recognizes human emotions through computer technology (*Ezzameli & Mahersia, 2023*). Emotional computing combines disciplines such as computer science, psychology, neuroscience, and linguistics to analyze and process various information such as human speech, facial expressions, and physiological indicators through computers, thereby inferring human emotional states and behaviors (*Guo & Lin, 2023*). In recent years, emotion recognition has received increasing attention due to its

applications in various research fields such as human-computer interaction, emotion analysis, user behavior detection, and fatigue monitoring (*Zhao et al., 2022*).

The methods of emotion recognition can be divided into two categories. One is based on physical signals, such as user language comments, facial expressions, gestures, *etc.*, which recognize the external representation of human emotions. Another approach is based on physiological signals, such as electroencephalogram (EEG), electrocardiogram (ECG), eye tracking (ET), *etc.*, to recognize the intrinsic representation of human emotions (*Gandhi et al., 2023*). *Zhao, Huang & Pan (2019)* used the word vector method to classify and recognize emotional tendencies by exploring user comments on products. *Kumar, Kishore & Pandey (2020)* classified human facial expressions using convolutional neural networks to achieve analysis and recognition of user emotions. These emotion recognition methods based on physical signals are simple to analyze and do not require much data information to obtain emotion recognition results. However, physical signals belong to human external reactions, and humans can hide their emotional expressions. In addition, image-based emotion recognition methods do not reflect genuine human emotions but instead, replace humans in emotion recognition of images (*Yang et al., 2023*). Therefore, the emotions conveyed by these physical signals cannot accurately capture the true thoughts of humans. Emotion recognition based on physiological signal methods such as EEG and eye tracking cannot be easily controlled by the subjects, as its objectivity and deep level of recognition have been proven to be a reliable emotion recognition method (*Khare et al., 2024*).

The intricate nature of human emotions renders single-modal data insufficient for a full-fledged emotional analysis (*Zhu et al., 2023*). This has led researchers to explore multimodal data combining various emotional indicators for more stable and comprehensive emotion recognition. Studies affirm that multimodal data offers a holistic view of emotional shifts, facilitating cross-verification among different data types (*Garg, Verma & Singh, 2023*). Current research predominantly integrates physiological data like EEG and eye movement recordings with behavioral data to formulate advanced emotion recognition systems. Such multimodal approaches have been shown to yield accurate insights into user emotions (*Khosla, Khandnor & Chand, 2020*; *Lim, Mountstephens & Teo, 2020*; *Jafari et al., 2023*).

For the experimental design of standard emotion recognition research, such as ET, EEG measurement, and other emotion cognition experiments, the central stimuli are images, videos, text, *etc.*, (*Babinet et al., 2022*; *Dzedzickis, Kaklauskas & Bucinskas, 2020*). During this process, participants observe emotional stimuli such as an image, a piece of text, or a movie clip and record the corresponding emotional responses while recording their physiological cognitive data. Classify these physiological cognitive data through corresponding emotional recognition frameworks and computational methods to identify the user's emotional cognition. However, emotional recognition is a collaborative process between the stimulus source and the individual, whereby the stimulus elicits the subject's emotional response. Although existing methods focus on the correlation of multimodal data, they do not consider the impact of stimulus sources on emotion recognition. They only use physiological data and ignore stimulus source data, which can lead to information loss in emotion recognition systems.

To compensate for the lack of information in emotion recognition systems, we propose a new framework for multimodal emotion recognition, which integrates stimulus source data, user behavior data, and cognitive physiological data to achieve emotion recognition. Figure 1 shows the multimodal emotion recognition framework. Case validation was conducted based on our proposed multimodal emotion recognition framework. We use a dataset of car images obtained from our team's previous research as stimulus samples (*Guo et al., 2023*), and use two types of emotional vocabulary as criteria (steady and lively) to obtain users' emotional responses to each car image as well as eye movement data and EEG data. Classify the multimodal data mentioned above to obtain user sentiment recognition. In addition, we will apply this framework to the existing publicly available dataset SEED (*Zheng & Lu, 2015*), and perform fusion calculations based on the stimulus source (video data) and EEG data of this dataset to determine the universality of the framework. Our innovations are as follows:

- Propose a new emotion recognition framework that integrates stimulus source data and physiological cognitive data to achieve emotion recognition.
- Propose a multimodal feature data fusion algorithm to adapt to this study's multimodal emotion recognition framework, which significantly improves accuracy compared to baseline methods.
- Given the limitation of the small sample size of the current physiological cognitive public dataset for emotion recognition, we will publicly disclose the data of this emotion cognition experiment.

Our contributions are twofold:

- We present a novel emotional recognition framework that integrates multiple data sources to improve emotional recognition accuracy.
- We propose a multimodal data fusion algorithm that surpasses the accuracy of traditional single-modal data or EEG and ET data fusion in emotional recognition.

The structure of the article is as follows. The second part introduces relevant work, including sentiment datasets, multimodal data fusion methods, sentiment recognition, *etc*. The third part introduces the methods of this article, mainly including experimental design, multimodal feature data extraction and fusion, data processing, and emotion recognition algorithms. The fourth part conducts case verification. The fifth part will be discussed. Finally, summarize this article.

## RELATED WORK

### Emotional dataset

In emotion recognition for users, data selection plays a crucial role. Stimulus data must be capable of eliciting distinctly differentiated emotional responses from users. For instance, the SEED dataset utilizes movie clips infused with positive, neutral, and negative emotions as stimulus data, and corresponding emotional physiological data is collected (*Zheng & Lu, 2015*). Similarly, the DEAP dataset employs music videos with varying arousal degrees as

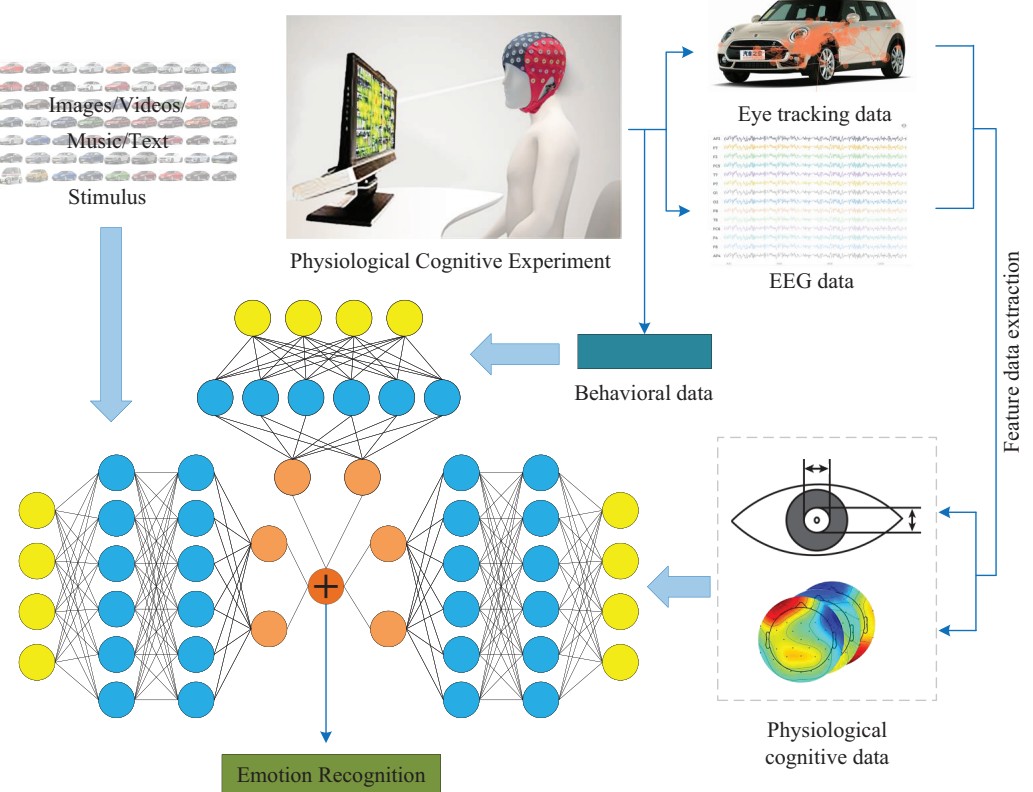

**Figure 1 Multimodal emotion recognition framework.** In this figure, the image of the stimulus component comes from the author's public data set. The car emotion image dataset can be accessed from the following website: https://doi.org/10.6084/m9.figshare.25484329.v1. Photo credit: Guo Zhuen. The presentation of data information components comes from pictures exported by BeGaze and EEGLab software. The remaining components come from Guo Zhuen's drawings.

stimulus samples, from which multimodal physiological data is also gathered (*Koelstra et al., 2012*). Standard stimuli used in emotion recognition datasets are illustrated in Table 1.

The sentiment dataset summarized in Table 1 cannot encompass all representative datasets, and related research has summarized more types of datasets (*Gandhi et al., 2023*; *Zhao et al., 2021*). Through understanding the existing publicly available datasets, it can be concluded that physical signal-based emotional cognition datasets include facial recognition, text, images, and more. The stimulus sources of physiological signal datasets are images, videos, texts, *etc.*, with various physiological signals, but mainly based on EEG and eye tracking data. However, the sample size based on physiological signals is relatively small. For example, the DEAP dataset with a large sample size is only 1,280, while the stimulus samples are only 40. Deep learning technology requires a large amount of data to support it and a sample size of data to support it. The sample size of existing emotional datasets based on physiological signals has limitations (*Zhang et al., 2022*). In addition, the emotion classification and recognition of known emotion datasets mainly involve intense emotional colors based on differentiation, such as positive, negative, happy, sad, *etc.*

**Table 1 Emotional recognition dataset.**

| Data set | Stimuli | Physical data | Physiological data | Classification | Samples |
|---|---|---|---|---|---|
| Aubt | Music | | EMG/ECG/SC/RSP | Joy/anger/sadness/pleasure | 100 |
| Multi-ZOL | | Text/image | | Physical score | 12,587 |
| SAVEE | Text/movie | Voice/FE | | Angry/happy/sad/neutral | 480 |
| IMDB | | Text | | Negative/positive | 50,000 |
| Sentiment140 | | Text | | Neutral/negative/positive | 1,600,000 |
| Twitter15 | | Text/image | | Neutral/negative/positive | 5,338 |
| Twitter17 | | Text/image | | Neutral/negative/positive | 5,972 |
| IAPS | | Image | | Valence/arousal/dominance | 956 |
| SEED | Movie | | EEG | Neutral/negative/positive | 15/625 |
| SEED-IV | Movie | | EEG/ET | Happy/sad/fear/neutral | 72/1,080 |
| DEAP | Music movie | | EEG/ECG/EMG/GSR | Valence/arousal/dominance | 40/1,280 |

**Note:**
EMG, electromyogram; SC, skin conductivity; RSP, respiration; GSR, galvanic skin response.

Although there has been basic research on emotion classification and recognition, which lays the foundation for future emotion recognition, it has not been able to deeply classify users' deep emotional cognition. Given this, we use our team's car image dataset as a stimulus source to construct a multimodal emotional cognition dataset and target deep emotional understanding in humans as a classification objective. Firstly, we plan to use 6,000 different types of car images as stimulus samples and use emotional vocabulary "steady" and "lively" as deep emotional evaluation information as emotional recognition targets. The subjects evaluated images of different types of cars based on emotional vocabulary evaluation goals, recorded eye tracking data, and EEG data. Finally, they obtained an emotion classification dataset with a large sample size, solving the limitations of small sample sizes in existing emotion datasets and the lack of in-depth exploration of human deep emotional understanding.

## Emotion recognition framework

Emotion recognition technology has many applications in natural language processing, social media analysis, market research, and user experience research. According to the category of processed data, it can be divided into physical signals and physiological signals, and according to the number of modes of data, it can be divided into single mode and multimodal. In recent years, sentiment classification and recognition have primarily relied on deep learning or machine learning methods for decision-making and inference (*Khare et al., 2024*). Therefore, we summarize common emotion recognition frameworks based on data information, modalities, and methods and compare them with the new multimodal emotion recognition framework proposed in this study. The common emotion recognition frameworks based on physiological cognitive data are shown in Table 2.

In emotion recognition based on physiological signal data, scholars mainly focus on single modal EEG data, multimodal EEG data, and eye movement data in addition to using the publicly available dataset DEAP multimodal data. In addition, the computational inference process mainly uses deep learning and machine learning techniques, such as

**Table 2 Emotion recognition frameworks based on physiological cognitive data with differences highlighted.**

| Reference | Data | Types | Methods | Difference |
|---|---|---|---|---|
| *Zhang et al. (2021)* | EEG/EMG/GSR, *etc.* | Multimodal | eDKMO | |
| *Pusarla, Singh & Tripathi (2022)* | EEG | Single | DCERNet | |
| *Moin et al. (2023)* | EEG/FE | Multimodal | SVM/KNN/Bagged | |
| *Xing et al. (2019)* | EEG | Single | SAE-LSTM-RNN | |
| *Liu & Fu (2021)* | EEG | Single | SVM | |
| *Jiang et al. (2020)* | EEG | Single | CNN | |
| *Guo & Lin (2023)* | EEG/ET | Multimodal | XGBoost | Physiological data |
| *Fei et al. (2023)* | EEG/ET | Multimodal | GAN | |
| *Wu et al. (2022)* | EEG | Single | CNN | |
| *Yin et al. (2017)* | EEG/EMG/GSR, *etc.* | Multimodal | MESAE | |
| *Yin et al. (2021)* | EEG | Single | GCNN-LSTM | |
| Linear regression | | | | |
| *Yang et al. (2020)* | EEG | Single | Linear regression | |
| Ours | Stimulus-EEG/ET | Multimodal | CNN-LSTM | Stimulus and physiological data |

CNN, SVM, LSTM, *etc.* A relatively novel emotion recognition calculation method, such as CNN-LSTM, is used to fuse different data. The difference between our research framework and existing emotion recognition frameworks is that we use multiple physiological signal data, integrate stimulus source data, and combine multiple deep learning algorithms for fusion calculation. Although traditional multimodal emotion recognition frameworks can effectively achieve emotion recognition, it is worth exploring whether combining two data types for emotion prediction can improve prediction accuracy. In addition, multimodal data fusion of stimulus source data can also explore issues such as reducing overfitting in physiological data prediction and improving the stability of prediction performance. Therefore, we propose integrating stimulus source data and stimulus source emotion induction data to achieve emotion recognition, a novel multimodal emotion recognition framework. This study will verify the robustness and accuracy of emotion prediction under this framework.

## Multimodal data fusion

As research progresses, it has been found that multimodal data fusion processing can be better used in emotional recognition (*Wang et al., 2020*), target detection (*Amsaprabhaa, Nancy Jane & Khanna Nehemiah, 2023*), image segmentation (*Farshi, Drake & Özcan, 2020*), and other applications. The method of multimodal data fusion has gained attention from researchers for its potential to improve classification performance. Three types of fusion methods can be categorized based on the level and method of fusion: feature-level, decision-level, and deep-learning hybrid (*Liu et al., 2022*). Feature-level fusion refers to the merging, addition, multiplication, and other operations of different modal features to form new multimodal features, followed by classification. The strength of this method is in utilizing information from different modalities to improve classification performance, with

an uncomplicated implementation and high computational efficiency (*Goshvarpour & Goshvarpour, 2023*; *Muhammad, Hussain & Aboalsamh, 2023*). Decision-level fusion combines the output results of different modal classifiers to form a new decision result using voting, weighted summation, or maximizing probability. The advantage of this method is in better utilizing the advantages of each modal classifier, reducing misjudgment and improving classification accuracy (*Zhang et al., 2023*; *Hu, Jing & Wu, 2023*). Based on deep learning hybrid fusion, different deep learning methods are used to process and extract features from data of different modalities, and fusion is carried out through feature-level fusion or decision-level fusion. When processing different modal data, methods such as deep neural networks, convolutional neural networks, recurrent neural networks, attention mechanisms, *etc.*, can be chosen, and the advantages of different fusion levels can be combined for selection. For example, *Pan et al. (2023)* proposed a method for assessing the alertness of high-speed train drivers based on physiological signals based on both decision-level and feature-level fusion. They first performed feature-level fusion calculations on EEG and ECG data using LSTM, filtering the result with a threshold to evaluate the driver's alertness. Similarly, *Akalya devi & Karthika Renuka (2023)* proposed a multimodal emotional recognition framework based on decision-level and feature-level fusion methods, using LSTM, CNN, and other methods to extract multimodal features and different fusion methods at different stages to achieve emotional recognition. Decision-level fusion and feature-level fusion are the foundations of deep learning hybrid fusion. Deep learning hybrid fusion can achieve the best results in multimodal data fusion by analyzing the characteristics of different modal data and processing them separately.

Based on the summary of the relevant work above, the existing research, including emotion dataset types, emotion recognition frameworks, and multimodal data fusion patterns, provides the foundation for this study. The foundation provided by the relevant work for this study and the limitations that this study aims to address are as follows:

1) The types of sentiment datasets in existing research are diverse and have been widely used and validated in sentiment recognition research. However, the number of stimulus samples in existing sentiment datasets is relatively small, which cannot meet the requirements of using a large amount of sample data in deep learning. Given this, this study will develop a multimodal emotion dataset suitable for emotion recognition, including stimulus source and physiological data, to provide a benchmark dataset for deep learning-based emotion recognition methods.

2) The emotion recognition framework has gradually evolved from single modality to multimodality and emotion recognition methods have evolved from linear classification to deep learning. Existing research provides a methodological foundation for conducting this study. However, existing emotion recognition frameworks mainly focus on identifying physical or physiological signals without effectively fusing stimulus source data. Therefore, this study proposes an emotion recognition framework that integrates stimulus source data and physiological data.

3) The research on multimodal data fusion methods is a hot topic in emotion recognition, and existing research provides a reference for data fusion methods in this study.

In this study, we propose a strategy based on deep learning hybrid fusion for emotional recognition and prediction of multimodal data. Firstly, we use feature-level fusion to extract EEG features through filtering and differential entropy methods. Based on the spatial distribution of EEG, we combine and structure the EEG features in a plane. This creates the EEG feature set. Next, we extract ET data features that can represent cognitive psychology, combine them, and construct the ET feature set. Secondly, using decision-level fusion, we apply CNN-LSTM to process the EEG feature set and obtain the output of EEG feature data. The ET feature set is then processed using DNN to output the ET feature data. We improve the VGGNet algorithm to obtain the output of stimulus source image features. Finally, decision fusion is again used to fuse the three categories of obtained multimodal feature data. User behavior data are combined, and deep neural networks process the fused data to achieve emotional recognition.

## METHODS

### Experimental design

To expand the emotion recognition framework proposed in this study to more fields, we summarized an experimental process based on traditional psychological cognition experiments (*Zheng & Lu, 2015*; *Koelstra et al., 2012*; *Zheng et al., 2019*; *Liu et al., 2022*). This process can obtain the subjects' behavioral data, physiological data, and stimulus source data. The experimental setup is as follows:

1) Selection of stimuli. According to the needs of emotional recognition, stimuli can be selected from images, videos, text, sound, *etc*., and multiple stimuli can also be combined.

2) Determine the output of emotional cognition. Based on the research objectives of emotion recognition of stimuli, determine participants' different emotional cognitive states towards stimuli, such as "positive neutral negative", "happy angry sad happy", or emotional vocabulary equivalence.

3) Emotional cognitive joint experiment. To obtain multimodal cognitive physiological data, multiple physiological measurement devices can be selected, such as EEG caps, ET devices, ECG devices, *etc*., to simultaneously measure the physiological and psychological states of subjects during the emotional cognitive process of stimuli and output multimodal data.

4) Obtaining cognitive and behavioral data of participants. After the subjects have an emotional understanding of stimuli, output indicators of emotional cognition appear. The subjects need to make quick choices based on cognitive understanding to obtain behavioral data such as their choices.

5) Multimodal data processing. Synchronize the physiological cognitive, behavioral, and stimulus source data of the subjects, and obtain feature data of each modality data.

The summary of emotional datasets found that the existing physiological signal-based emotional datasets have a small sample size. At the same time, deep learning techniques require a large amount of sample data. Therefore, based on the proposed experimental

process, cognitive experiments were conducted with publicly available team datasets. Therefore, this study uses car images as stimulus samples to analyze the emotional perception of participants. Compared with traditional emotions such as "positive neutral negative" and "happy angry sad happy", we use the emotional adjective "steady lively" with high-level emotional understanding as emotional recognition information. Combining common physiological cognitive data, we use EEG caps and ET devices to measure the physiological data of users. The specific experimental process is shown in Fig. 2.

## Data extraction

Users' physiological and behavioral data can be acquired based on the emotional recognition framework. As feature extraction is unnecessary for behavioral data, the multi-modal feature data focuses solely on the physiological and stimulus source data. Figure 3 illustrates the processing methodology of multi-modal feature data. Its main components are the EEG feature data extraction, ET feature data extraction, and stimulus source data extraction. To integrate multi-modal data, it is essential to synchronize the data information. Different data trails can be divided based on the stimulus source, and behavioral labels for each trail can be marked. For EEG data, it is necessary to construct a multidimensional feature data structure to integrate the frequency, spatial, and temporal characteristics of EEG signals. Because the stimuli in each experiment are different, we need to conduct a single trial analysis for each stimulus. The differential entropy feature has been proven to be a reliable method for analyzing single-trial EEG signals and a stable method for emotion recognition in related research. This study will use differential entropy (DE) as the feature extracted from EEG data. Firstly, we decompose each trial segment into five frequency bands using a Butterworth filter, including $\delta(1\ \text{Hz} \sim 3\ \text{Hz})$, $\theta(4\ \text{Hz} \sim 7\ \text{Hz})$, $\alpha(8\ \text{Hz} \sim 13\ \text{Hz})$, $\beta(14\ \text{Hz} \sim 30\ \text{Hz})$ and $\gamma(30\ \text{Hz}\sim)$. The purpose of doing this is to integrate frequency information from different EEG data. Subsequently, we extracted differential entropy features for each segment band using 0.5 s as a window to obtain the temporal characteristics of changes in EEG signals during the emotional cognition of the subjects.

The differential entropy feature $H(x)$ of continuous EEG data $X$ can be defined as:

$$H(x) = \int_{-\infty}^{\infty} p(x) \log(p(x)) dx \tag{1}$$

where $p(x)$ is the probability density function of $X$.

When defining the probability density function $p(x)$ of continuous EEG data $X$, it is expected to employ statistical methods or data modeling techniques. Here, we utilize the Gaussian distribution to describe the probability density function $p(x)$ of continuous EEG data $X$. The probability density function of the Gaussian distribution is given by:

$$p(x) = \frac{1}{\sqrt{2\pi\sigma^2}} \exp\left(-\frac{(x-\mu)^2}{2\sigma^2}\right) \tag{2}$$

where $\mu$ represents the mean and $\sigma$ denotes the standard deviation of the Gaussian distribution. Finally, according to the distribution of different electrode channels, the

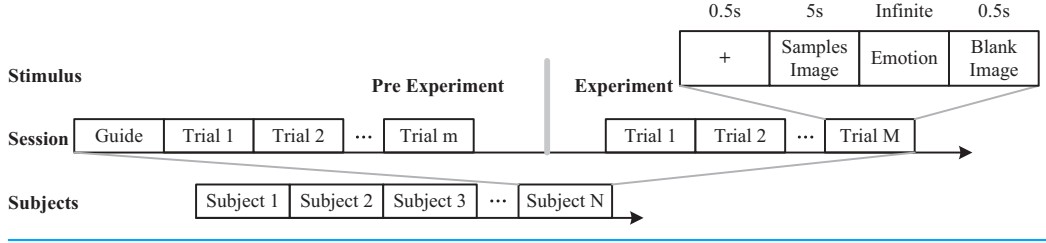

**Figure 2 Experimental process.** Image credit: Zhuen Guo.

**Figure 3 Multimodal feature data processing.** In this figure, the image of the stimulus component comes from the author's public data set. The car emotion image dataset can be accessed from the following website: https://doi.org/10.6084/m9.figshare.25484329.v1, and the presentation of data information components comes from pictures exported by BeGaze software and EEGLab software, and the remaining components come from Guo Zhuen's drawings.

extracted differential entropy features are stacked in two dimensions using the feature level fusion method to obtain the spatial distribution characteristics of EEG signals. In this article, we set up an 11 * 11 two-dimensional space to integrate the distribution positions of different electrode channels. As shown in Fig. 4, the electrode and two-dimensional plane information of the EEG cap we used are shown respectively, where 0 represents no data at that position.

Regarding ET data, we conducted a separate feature extraction of the left and right eyes. It has been demonstrated in related research that ET features, such as pupil diameter, number of fixations, and saccade count, can reflect the emotional and cognitive state of the user (Guo et al., 2019). We undertook data filtering and selection based on existing studies and the data obtained from each trial, culminating in the acquisition of 52 distinctive feature categories (Mele & Federici, 2012), as illustrated in Table 3. We employed a feature-level fusion method to merge the ET feature data to obtain an ET feature dataset.

We resized images evenly and extracted unique multi-color features to obtain stimulus source features. Consequently, we acquired a feature set of size $224 \times 224 \times 3$.

## Multimodal fusion network

We propose an Emotion-Multimodal Fusion Neural Network (E-MFNN), a novel multimodal fusion neural network based on our new emotional recognition framework. E-MFNN synchronously fuses physiological cognitive data, user behavioral data, and stimulus source data to extract the characteristics of each modality and forecast the emotional state. E-MFNN incorporates decision-level and feature-level fusions. In the beginning, the decision-level fusion is utilized to process and train the data of each modality by different neural networks to acquire the main characteristics of each respective modality's data. Next, based on the feature-level fusion technique, the aforementioned main characteristics of each modality's data are concatenated to generate a characteristic data set containing each modality of the data. Finally, the final dataset produced by integrating feature sets of each modality's data is trained to provide predicted values. The E-MFNN network structure is displayed in Fig. 5.

For EEG data, each trial includes multiple datasets of differential entropy features based on the time series due to the extraction of differential entropy features. A CNN network extracted the EEG's frequency and spatial information. Still, since the spatial information dimension of EEG signals is relatively small, we did not increase the pooling layer when using CNN to preserve more feature information. Our neural network consists of eight convolutional layers with a ReLu activation function, and the last convolutional layer is connected to three fully connected layers, which ultimately output 256 features. We used an LSTM artificial neural network to process all output feature data from the trial in combination with the same CNN model for handling other differential entropy feature data of the time series. Finally, we obtained EEG feature output that corresponded to each trial stimulus.

To process ET data, the data size needs to be transformed from $52 \times 2$ to $52 \times 2 \times 1$ before being inputted into the CNN module explicitly designed for ET data processing. In contrast to the CNN module for EEG feature extraction, this one lacks the LSTM process

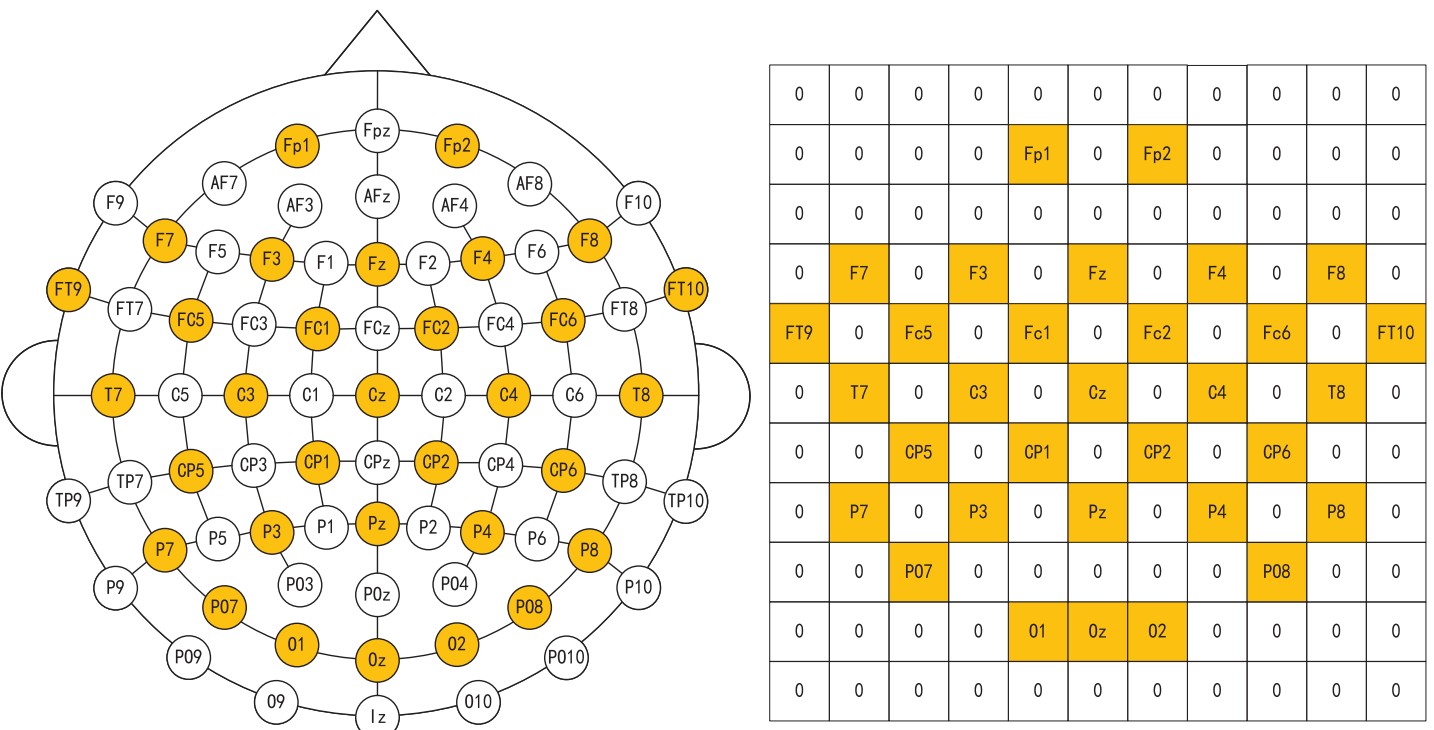

**Figure 4 Electrodes of EEG caps and corresponding two-dimensional maps.** Image credit: Zhuen Guo.

for feature data extraction. The VGGNet16 neural network architecture was utilized for stimulus source data, as it remains the most prevalent architecture for image classification despite being proposed earlier (*Simonyan & Zisserman, 2015*). Unlike the VGGNet16 architecture, the CNN module for processing stimulus source data generates a feature map size of 256 as the output of its last fully connected layer without applying the Softmax calculation. Each modality dataset produces 256 features following processing through different neural networks and concatenating these features through the feature-level fusion method. Training these features, three fully connected layers, and the Softmax function can ultimately make emotional response predictions. The configuration of the E-MFNN structure is displayed in Table 4.

# EXPERIMENTAL VERIFICATION AND RESULTS

## Emotional cognition experiment

### Preparation of stimulus samples

As elaborated in "Emotional dataset", this study employed specimens of cars as stimuli and utilized a car dataset of our emotional images as the source of stimuli. The dataset comprises 25,152 pictures of various car models and colors from orthogonal perspectives, captured against a white background to downplay the effect of extraneous variables on emotional components. Based on the emotional data obtained previously, we categorized 6,000 car images to form the stimulus sample pool, selecting 3,000 images randomly for

**Table 3 ET features indicators.**

| ET features | |
|---|---|
| The total of stimulus selection time (ms) | Saccade amplitude maximum (°) |
| Blink count | Saccade amplitude minimum (°) |
| Blink frequency (count/s) | Saccade velocity total (°/s) |
| Blink duration total (ms) | Saccade velocity average (°/s) |
| Blink duration average (ms) | Saccade velocity maximum (°/s) |
| Blink duration maximum (ms) | Saccade velocity minimum (°/s) |
| Blink duration minimum (ms) | Saccade latency average (ms) |
| Fixation count | Entry time (ms) |
| Fixation frequency (count/s) | Sequence |
| Fixation duration total (ms) | Net dwell time (ms) |
| Fixation duration average (ms) | Dwell time (ms) |
| Fixation duration maximum (ms) | Glance duration (ms) |
| Fixation duration minimum (ms) | Diversion duration (ms) |
| Fixation dispersion total (px) | First fixation duration (ms) |
| Fixation dispersion average (px) | Glances count |
| Fixation dispersion maximum (px) | Revisits |
| Fixation dispersion minimum (px) | Fixation count |
| Scanpath length (px) | Visible time (ms) |
| Saccade count | Net dwell time (%) |
| Saccade frequency (count/s) | Dwell time (%) |
| Saccade duration total (ms) | Fixation time (ms) |
| Saccade duration average (ms) | Fixation time (%) |
| Saccade duration maximum (ms) | Average pupil size |
| Saccade duration minimum (ms) | Minimum pupil size |
| Saccade amplitude total (°) | Maximum pupil size |
| Saccade amplitude average (°) | Variance of pupil size |

each emotion class (steady and lively), of which the two categories had approximately the same number of samples.

### Recruiting participants

To delve into participants' emotional responses towards products while nullifying the influence of individuals with specific emotional recognition linked to product design, car salespeople, *etc.*, we recruited test subjects from Guizhou University and remunerated them suitably. With an age range of 21, 28, 30 participants were the final selection for the emotional recognition experiment, with 16 males and 14 females being right-handed, having normal vision or corrected vision, and having no history of mental illness or related conditions. Participants received training on the purpose of the experiment and possible challenges that may arise from the test.

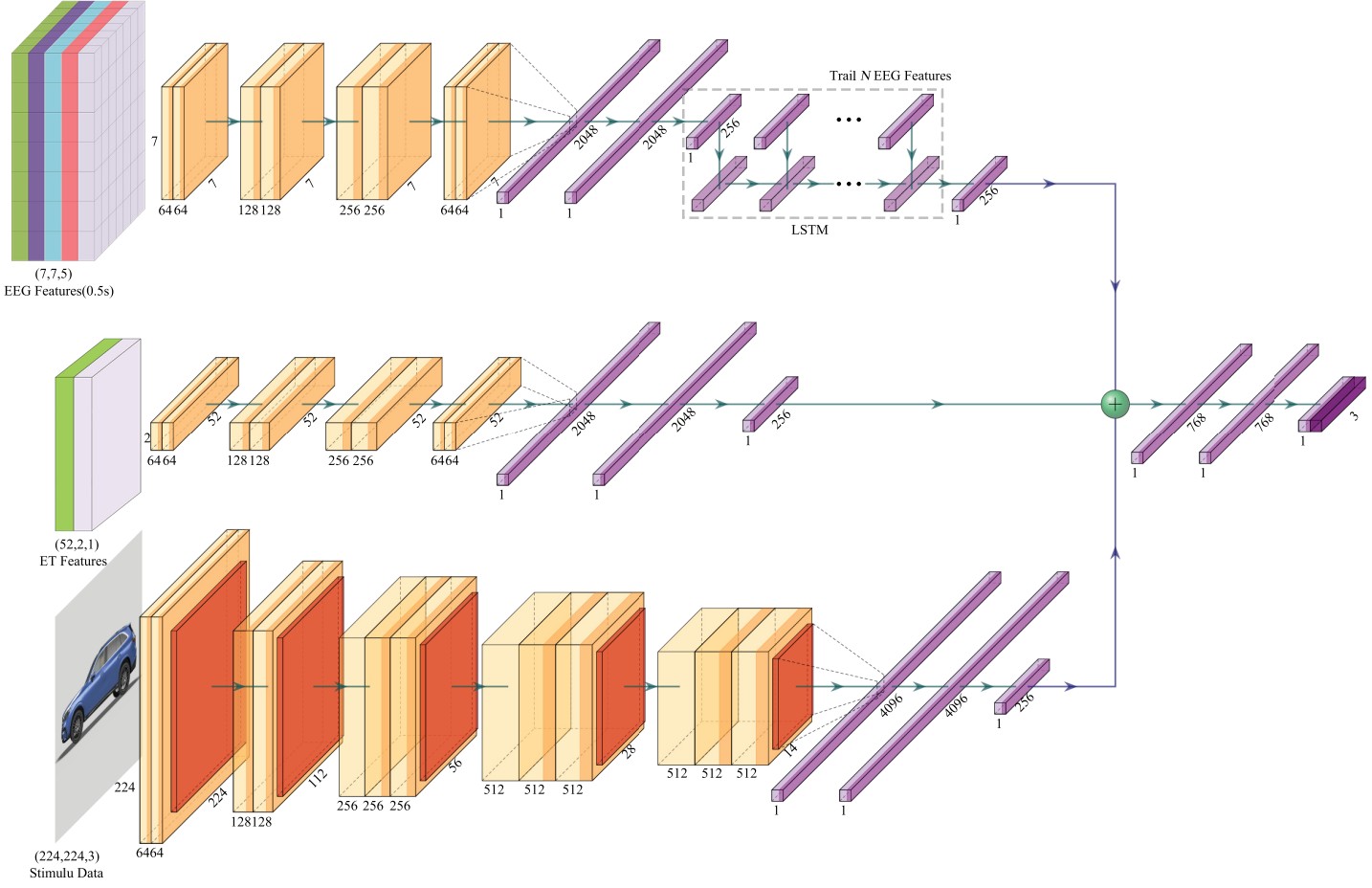

**Figure 5 The neural network architecture of E-MFNN.** In this figure, the image of the stimulus component comes from the author's public data set. The car emotion image dataset can be accessed from the following website: https://doi.org/10.6084/m9.figshare.25484329.v1. Image credit: Guo Zhuen.

## Experimental process

The emotional-cognitive experiments were conducted using a range of equipment. One of them is the EPOC Flex EEG cap, which includes 32 electrode channels and has a resolution of 128 Hz; the SMI RED500 desktop eye tracker, which has a resolution of 120 Hz; a stimulus presentation computer; and separate EEG and ET data recording computers. During the experiments, E-Prime two Psychology Experiment software is used to integrate these devices and present sample stimuli. EEG data and information are recorded using Emotiv Pro, while the iView X and BeGaze tools record and process ET data. The operating environment relied upon during these studies is illustrated in Fig. 6.

A total of 6,000 stimulus samples were randomly distributed among 60 groups, each consisting of 100 samples. The experimental area was maintained at a consistent brightness level and was free of extraneous sound or lighting interference. Before experimenting, participants were required to wash their hair to help reduce the influence of scalp oil on conductivity. During the experiment, a pre-test was performed to allow

**Table 4 Configuration of neural network structure for E-MFNN.**

| E-MFNN | | | | |
|---|---|---|---|---|
| **EEG data** | | | **ET data** | **Image** |
| **DE data** | ... | **DE data** | | |
| Input (11 * 11 * 5) | ... | Input (11 * 11 * 5) | Input (52 * 2 * 1) | Input (224 * 224 * 3) |
| conv3-64 | | conv3-64 | conv3-64 | conv3-64 |
| conv3-64 | | conv3-64 | conv3-64 | conv3-64 |
| conv3-128 | | conv3-128 | conv3-128 | Maxpool |
| conv3-128 | | conv3-128 | conv3-128 | conv3-128 |
| conv3-256 | | conv3-256 | conv3-256 | conv3-128 |
| conv3-256 | | conv3-256 | conv3-256 | Maxpool |
| conv3-64 | | conv3-64 | conv3-64 | conv3-256 |
| conv3-64 | | conv3-64 | conv3-64 | conv3-256 |
| FC-2048 | | FC-2048 | FC-2048 | conv3-256 |
| FC-2048 | | FC-2048 | FC-2048 | Maxpool |
| FC-256 | | FC-256 | | conv3-512 |
| | | | | conv3-512 |
| | | | | conv3-512 |
| | | | | Maxpool |
| | | | | conv3-512 |
| | | | | conv3-512 |
| | | | | conv3-512 |
| | | | | Maxpool |
| | | | | FC-4096 |
| LSTM | | | | FC-4096 |
| 256 | | | FC-256 | FC-256 |
| Concatenate | | | | |
| FC-768 | | | | |
| FC-768 | | | | |
| FC-10 | | | | |
| Softmax | | | | |

individuals to become familiar with the operating requirements. The formal experiment included the appearance of a "+" symbol, displayed for 0.5s, to draw participants' attention. This was followed by a 5 s presentation of a stimulus sample image, upon which participants were shown a set of emotional words, for example, sedate and lively. Participants were then required to choose one of these options. Although the choice had no set time limit, the reaction time was recorded. A blank image was shown for 0.5 s after each trial to remove any visual afterimage. Each subject participated in two experiments at different intervals, requiring roughly 25 min of preparatory time.

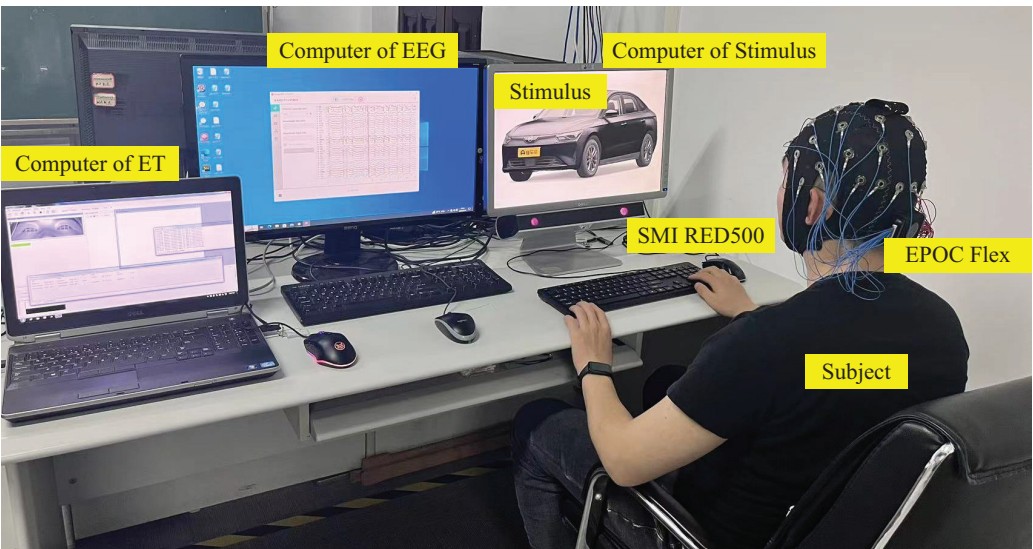

**Figure 6 Test operating environment.** Photo credit: Zhuen Guo.

## Data processing

Firstly, we processed the physiological and behavioral data of the subjects by screening the collected data to improve the accuracy of our study. We detected incomplete EEG data in two sets of experiments, covering 21 Trails, and incomplete ET data in one set of experiments, covering six Trails. We eliminated the incomplete data, acquiring a complete dataset containing 5,973 sets of physiological data, user behavior data, and stimulus source data.

Subsequently, we proceeded to process the obtained data. To preprocess the EEG data, we utilized the EEGLAB toolbox. The processing included importing the data and stimulus label, locating the electrodes, re-referencing using the average reference method, filtering, cropping the Trail time, performing baseline correction, performing Independent Component Analysis (ICA), and removing artifacts. This resulted in the acquisition of 5,973 segments of EEG data containing 32 electrodes.

Finally, we extracted EEG feature data using filtering and differential entropy methods, acquiring 5,973 sets of differential entropy containing 10 differential entropy features with time series for 32 channels and five frequency bands. The EEG data was formed into a $(5,973 \times 32 \times 5 \times 10)$ dimensional dataset. We extracted ET features as per "Data extraction" and calculated the values of each ET feature indicator. A total of 52 features were obtained for the left and right eyes, resulting in a $(5,973 \times 2 \times 52)$ dimensional ET dataset. We reduced the image resolution to $224 \times 224$ pixels for the stimulus source data and normalized the data. This process resulted in a $(5,973 \times 224 \times 224 \times 3)$ dimensional stimulus source dataset.

Finally, the preprocessed data was normalized and transformed into datasets. The datasets were divided into training, validation, and test sets at an 8:1:1 ratio and were used to train and validate the E-MFNN model. To enable efficient training, the State Key

Laboratory of Public Big Data, Guizhou University, provided server clusters with an Ubuntu 18.04 environment, using the TensorFlow framework and a single NVIDIA A100 processor. In addition, we also used different modal data combinations and neural network architectures for ablation experiments to verify the effectiveness and advantages of the emotion recognition framework and E-MFNN proposed in this study. The combination methods of data include the fusion of EEG-ET data, EEG data, stimulus source data, ET data, EEG data fusion stimulus source data, and ET fusion stimulus source data. The training network is a variation of the E-MFNN proposed in this study, which involves retaining the network module that trains the data and deleting the training module that did not input the data.

## Experimental result

We will train each data type for 100 epochs and update the training parameters using the SGD optimizer. In addition, to ensure comparison with existing technologies and fully validate the performance of the multimodal emotion recognition framework proposed in this study, we compared our results with the latest emotion recognition techniques based on EEG physiological data.

Standard emotion recognition methods based on EEG data and deep learning.

1D-CNN (*Aldawsari, Al-Ahmadi & Muhammad, 2023*). In emotion recognition based on EEG data, EEG signals are usually one-dimensional. The convolutional layers of one-dimensional CNN architecture are used to process EEG physiological data, achieving emotion recognition and classification. The baseline validation will be trained using a 1D-CNN network, and comparative experiments will be conducted by normalizing and one-dimensional stretching the EEG data of each stimulus trail.

2D-CNN (*Farokhah, Sarno & Fatichah, 2023*). 2D-CNN is the default architecture of CNN, which converts one-dimensional EEG signals into two-dimensional structures through preprocessing techniques and uses 2D-CNN for sentiment recognition to achieve classification. During baseline validation, 32 electrode channels will be arranged in a two-dimensional manner, and each electrode channel will be a one-dimensional EEG sequence data.

3D-CNN (*Yuvaraj et al., 2023*). By defining the spatial layout of EEG data, the three-dimensional structure of EEG data can be visualized and expressed. Data from different bands of each electrode channel will be extracted during baseline validation, and the electrode channels will be laid out to form a three-dimensional data structure.

4D-CNN (*Shen et al., 2020*). Four-dimensional structured data is formed by defining the spatial layout of EEG data and extracting relevant data features based on time series. During baseline validation, the band data of each electrode channel is extracted, and the differential entropy features of the data are extracted to form four-dimensional structural data.

RNN (*Liu, Su & Liu, 2018*). EEG data is physiological data of time series, which is trained through RNN recurrent networks and extended network structures such as LSTM, BERT, and transformer to achieve emotion recognition. During baseline validation, LSTM

**Table 5 The results of each framework.**

| Number | Type | Data | Model | Accuracy | Training speed (it/s) | Test speed (it/s) |
|---|---|---|---|---|---|---|
| 1 | Baselines | EEG | 1D-CNN | 71.80% | 7.39 | 17.12 |
| 2 | | EEG | 2D-CNN | 73.51% | 7.14 | 17.05 |
| 3 | | EEG | 3D-CNN | 86.13% | 7.81 | 16.26 |
| 4 | | EEG | 4D-CNN | 85.86% | 6.95 | 14.54 |
| 5 | | EEG | LSTM | 82.65% | 6.12 | 15.17 |
| 6 | Ablation experiment | ET | | 63.57% | 8.16 | 20.18 |
| 7 | | EEG | | 85.06% | 7.11 | 16.35 |
| 8 | | Stimulus | | 87.36% | 9.36 | 22.51 |
| 9 | | EEG-ET | | 86.95% | 6.31 | 13.49 |
| 10 | | EEG-Stimulus | | 89.03% | 6.16 | 14.51 |
| 11 | | ET-Stimulus | | 87.25% | 5.95 | 14.3 |
| 12 | Our | Multimodal | E-MFNN | 93.85% | 5.31 | 13.32 |

networks are used to process the EEG data of each electrode channel separately. Finally, the features of different electrode channels are fused to achieve emotion recognition.

Table 5 showcases the multimodal emotion recognition framework, the baseline method introduced in this study, and the results from ablation experiments conducted for each data modality.

Through comparison with baseline methods and ablation experiments, we found that our proposed multimodal emotion recognition framework has significant advantages. However, the sentiment classification accuracy of EEG data in baseline methods can reach over 85%, such as through 3D-CNN and 4D-CNN models. However, our framework has significantly improved performance, reaching 93.85%. In addition, the results of the ablation experiment also indicate that emotion recognition based on eye movement data is ineffective. Although some models based on baseline methods have advantages in emotion recognition of EEG data compared to emotion recognition only targeting EEG data in our ablation experiments, we did not integrate the baseline method model with the multimodal framework model. The main reason is that the baseline method model is relatively complex. If the baseline method model is fused with our model, the training speed is slower, and higher servers must be configured for training.

Based on the Table 5, we compared the training and testing speeds of different models with our model, using the same batch size (64 as shown in the table). The results indicate that although our model exhibits superior accuracy compared to other models, it lags in both training and testing speeds. Despite having a significantly larger model size and dataset than other models, the difference in training and testing speeds is not substantial. In terms of time cost, the discrepancy is negligible.

To better analyze the prediction of emotion recognition using different data, we selected representative EEG-ET fusion data and EEG data related to changes during the training process. Figure 7 summarizes the fitting curves of the accuracy of using multimodal data, EEG-ET fusion data, and EEG data as a function of the training cycle and includes the

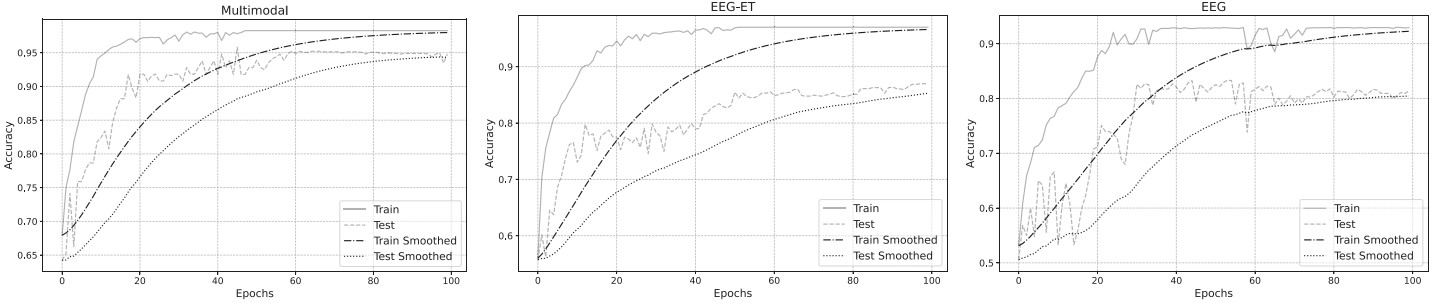

**Figure 7 Accuracy of emotion recognition for different data types.**

fitting curves of the training and testing sets and the comparison process. Our novel discovery is that for our research goal, the binary classification problem of emotion recognition, the accuracy should ideally exceed 95%. We can see from the comparison of training results with different data that the classification performance of the multimodal emotion recognition framework proposed in this study gradually increases with the increase of the training cycle, and the recognition accuracy will eventually reach over 90%. The accuracy of emotion recognition training sets based on traditional methods that only target EEG-ET fusion data to obtain EEG data can reach over 90%. Still, the accuracy of the test set differs significantly from that of the training set. For example, the accuracy of the test set for EEG-ET fusion data can only reach over 85%, while the accuracy of emotion recognition based on EEG data is only about 80%. In traditional emotion recognition methods, the accuracy of the test set fluctuates wildly. In contrast, the accuracy of the validation and test sets of this study's multimodal emotion recognition framework fluctuates less with the training cycle, resulting in a more stable emotion recognition effect.

To delve deeper into the emotion recognition performance across three data types, we analyzed their training loss functions during the training process. Figure 8 illustrates the fitted loss value curves for multimodal, EEG-ET fusion, and EEG data, plotted against the training duration. These curves reveal that irrespective of the data type, the loss values of the training set consistently diminish as training progresses. Notably, the training loss for the multimodal emotion recognition framework introduced in this study exhibits a steady decline over time. In contrast, the loss values for emotion recognition using EEG-ET fusion and EEG data show more pronounced fluctuations. This comparison underscores the stability of our proposed multimodal framework in emotion recognition, demonstrating its potential to mitigate overfitting to a certain degree.

The variations in accuracy and loss values indicate that while traditional physiological data has some applicability in emotion recognition, its stability is somewhat lacking, and the quality and quantity of the data are critical factors. The multimodal emotion recognition framework proposed in our study integrates stimulus source data, effectively reducing data overfitting and enhancing the stability of emotion recognition. To investigate the effectiveness of emotion recognition across different data types further, we analyzed the accuracy changes among the three data types. Figure 9 summarizes the

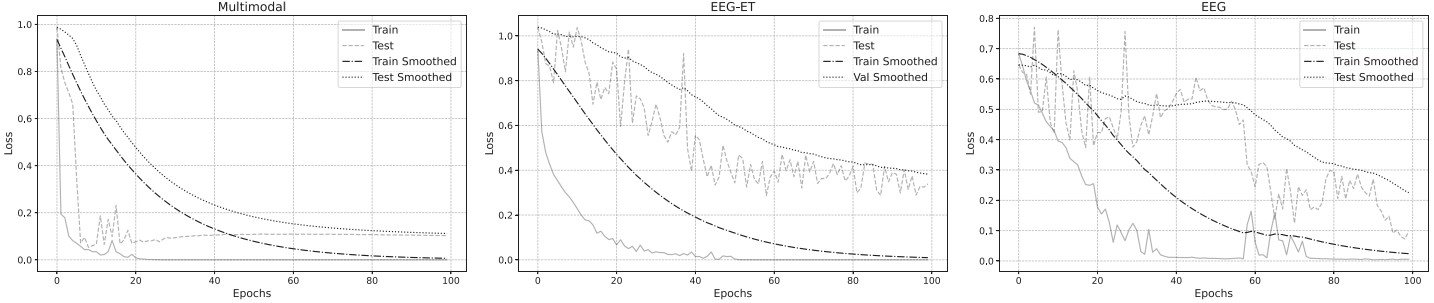

**Figure 8 Loss of emotion recognition for different data types.**

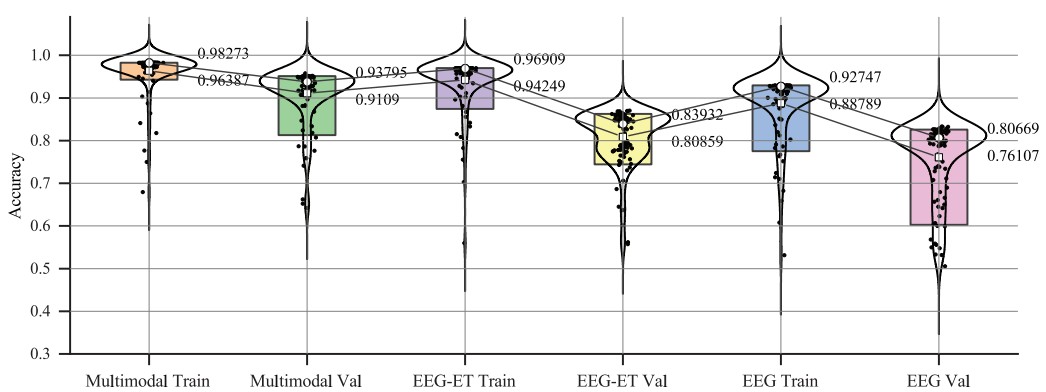

**Figure 9 Accuracy distribution of emotion recognition for different data types.**

accuracy data for each type during the training and validation phases. The figure illustrates that within a 5% to 95% confidence interval, the accuracy distribution for emotion recognition using multimodal data exhibits a tighter clustering than that of EEG-ET and EEG data, signifying a more consistent and reliable performance in emotion recognition.

## Evaluation

To assess the efficacy of various data types and network models, we employed test data to compare the classification performance of different modal data types, utilizing confusion matrices as our analytical tool. Figure 10 displays the emotion recognition prediction results achieved through multimodal and stimulus source data. In this comparative analysis, the emotion recognition framework newly proposed in our study achieved a prediction accuracy of 94.53%. In contrast, the prediction accuracy for emotion recognition utilizing EEG-ET fusion data stood at 89.5%, while the prediction accuracy based solely on EEG data was 87.33%. These results demonstrate that the emotion recognition predictions derived from multimodal data significantly outperform those based on stimulus source data alone, highlighting our proposed framework's enhanced accuracy and reliability.

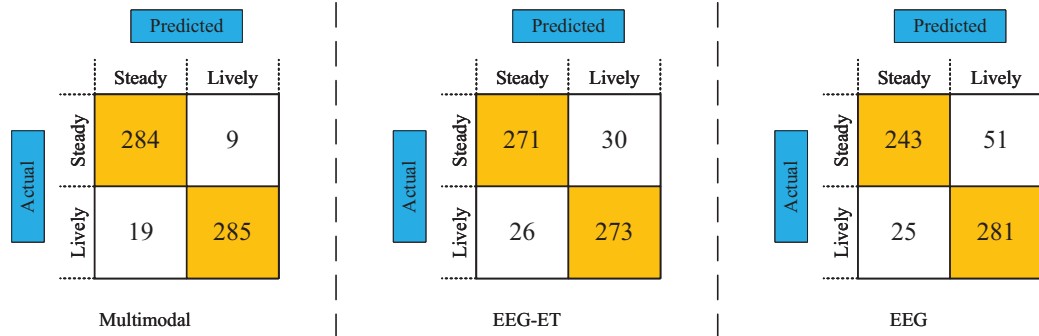

**Figure 10 Confusion matrix of emotion recognition for different data types.**

From the comparison results of different methods, our proposed multimodal emotion recognition framework performs better than traditional methods based on EEG-ET fusion emotion recognition or analyzing EEG data for emotion recognition. The primary manifestation of our proposed emotion recognition framework is based on multiple types of data for emotion recognition, including physiological data of subjects, such as eye tracking data and EEG data, behavioral data of subjects, and stimulus source data. By integrating multiple data types, we can obtain more data features generated by the emotional understanding process of subjects toward stimuli. In addition, our proposed E-MFNN also plays a vital role in emotion prediction. We process data types separately and fuse them through decision and feature fusion to obtain excellent prediction results.

## DISCUSSION

This study robustly demonstrates the viability of multimodal data fusion in emotion recognition through extensive experimentation. And propose E-MFNN to make it more suitable for emotion recognition and prediction in multimodal data. The results in "Experimental Verification and Results", when compared with other modal data-based emotion recognition methods, affirm the superiority of our proposed multimodal emotion recognition framework. This framework exhibits several critical advantages over traditional approaches: (1) It achieves higher prediction accuracy; (2) It is less prone to overfitting during the training process; (3) It enables participants to discern deeper emotional understandings of stimuli beyond merely recognizing emotions associated with overt emotional expressions.

Firstly, compared with other studies, the EEG-ET approach shows lower effectiveness in emotion recognition and prediction. This study's EEG and ET feature data are based on relevant works. The EEG data feature extraction is similar to previous studies, which utilized differential entropy features and conducted a spatial feature transformation (*Shen et al., 2020*). However, they used a longer single EEG data duration (4 min), while our study had only 5 s. The extraction of ET feature data is also similar to prior studies, which extracted multiple ET features (*Liu, Zheng & Lu, 2016*). However, we extracted more ET feature indicators than prior studies, utilizing 52 ET feature indicators separated by left and right ET. Thus, we suspect the poor performance of emotion recognition after eye-brain

data fusion may relate to the stimulus duration and feature selection during training. Furthermore, selecting more ET feature indicators may affect prediction accuracy. Hence, future research can focus on obtaining more representative feature indicators in the case of shorter single test durations. This issue should also be considered in emotion recognition studies that incorporate EEG-ET. Despite the noteworthy performance of our proposed multimodal emotion recognition framework, addressing these issues will further increase our emotion recognition accuracy and robustness.

Next, the contribution of E-MFNN to the recognition and prediction of emotions in multimodal data is a noteworthy achievement that has produced excellent results. However, E-MFNN still has room for improvement to accommodate a broader range of multimodal fusion data. This study's proposed emotion recognition framework can obtain physiological cognitive data, including electroencephalogram and ET data. E-MFNN processes electroencephalogram data based on time series while directly processing ET data through the CNN module. Scholars have shown that ET data fully reflects the subject's cognitive process (*Zhu et al., 2022*). Yet, we only utilized the cognitive results of ET data in the study, neglecting the extraction of cognitive feature indicators corresponding with alterations in ET data. Therefore, further improvement of E-MFNN can be achieved by adding time-based data processing while processing ET data. This adjustment will supplement the subject's cognitive feature extraction and promote a more profound comprehension of emotions in the prediction results. E-MFNN is an exemplary algorithm for processing multimodal emotional data, which can be modified to handle different data types by introducing additional processing modules and fusing the data *via* feature and decision-level fusion training. These modifications will improve its applicability to a broader scope of emotional data processing.

Introducing this new multimodal emotion recognition framework aims to expand its applicability across a broader spectrum of emotion recognition scenarios. In this study, we utilized physiological and stimulus source data from emotional cognition experiments for emotion recognition. To assess the framework's versatility, we tested it against the publicly available SEED dataset, a three-class dataset comprising EEG physiological data (*Zheng & Lu, 2015*). The SEED dataset employs movie clips as stimuli to elicit emotional responses and EEG data from participants. Although SEED doesn't provide direct stimulus source data, it offers comprehensive details about the stimuli, including video source websites and time segment information.

Leveraging our multimodal emotion recognition framework, we integrated video and EEG data to facilitate emotion recognition inference. We extracted movie segment information and derived a stimulus source dataset with a time series by capturing one frame per second from the segments. Given that the dataset has only 15 stimulus fragments but 625 sample instances, we applied data augmentation techniques, such as cropping and rotation, to the frame data. LSTM networks were used to train the time-series frame images, which were then fused with LSTM networks processing the EEG data. A CNN was employed to handle the features from both sources, culminating in the output of emotion recognition. The process and methodology are depicted in Fig. 11.
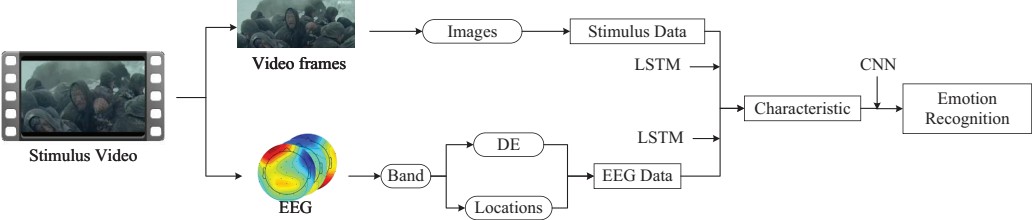

**Figure 11 The application of multimodal emotion recognition framework on SEED dataset.** The presentation of data information components comes from pictures exported by EEGLab software. The remaining components come from Guo Zhuen's drawings.

**Table 6 Comparison of emotional recognition accuracy based on SEED dataset.**

| Number | Author | Accuracy |
|---|---|---|
| 1 | *Delvigne et al. (2022)* | 84.11/2.9 |
| 2 | *Li et al. (2021)* | 93.12/6.06 |
| 3 | *Li et al. (2018)* | 92.38/7.04 |
| 4 | *Shen et al. (2020)* | 94.74/2.32 |
| 5 | Our | 96.78/3.51 |

Our study undertook a comparative analysis with the latest research in emotion recognition using the SEED dataset, employing a five-fold cross-validation method to ascertain recognition accuracy. This comparison, detailed in the accompanying Table 6, demonstrates that our proposed emotion recognition framework surpasses existing state-of-the-art models in emotion recognition accuracy within the SEED datasets. By incorporating stimulus sources (video data), our framework achieved a remarkable accuracy rate of 96.78%, effectively inferring users' emotional states. This outcome not only validates the effectiveness of our framework but also attests to its universality. It underscores the potential of our proposed multimodal emotion recognition framework to be applicable in diverse emotion recognition scenarios involving both stimulus sources and physiological data.

However, our study did not extend the verification to other publicly available datasets like DEAP and SEED-IV, primarily due to their lack of stimulus source data. Despite efforts to locate such data based on information provided in these public datasets, including website sources and descriptions, we encountered significant challenges. A substantial portion of the stimulus source data was either inaccessible or unplayable. As a result, our validation was confined to the SEED dataset. Future research endeavors will continue to explore our framework's applicability to a broader range of datasets. This expansion will enhance our understanding of the framework's versatility and effectiveness in various emotion recognition contexts.

## CONCLUSION

In this research, we developed an innovative emotion recognition framework that synergizes stimulus source, behavioral, and physiological cognitive data. To effectively integrate and train multimodal data for emotion recognition, we introduced the E-MFNN algorithm. This framework and the E-MFNN demonstrate enhanced accuracy and robustness in emotion recognition compared to traditional methods and data types, such as EEG-ET fusion data and stimulus source data. A key aspect of our study involved using car images, which necessitated a deep emotional understanding from participants as experimental cases. We captured physiological cognitive data through ET devices and EEG caps. However, it is important to note that this is merely a case study validation. The crux of our research lies in the proposed framework's ability to recognize emotions in response to diverse stimuli and employ various physiological measurement techniques, showcasing its broad adaptability.

In addition, as previously noted throughout this study, our experimental design utilized emotionally evocative images from past research to evaluate subjects' emotional comprehension at a deeper level, as opposed to stimuli with easily distinguishable emotional expressions. The results were deemed satisfactory regarding emotion recognition and prediction. However, we feel there is a lack of significant difference between our results *vs* those presented by stimuli with clear emotional distinctions. As emotion recognition is the product of classifying induced data, the resulting data is devoid of deep emotional processing based on the data. As a result, we are motivated to investigate further whether different types of modality data correlate during emotion recognition training and how they differ from data generated by stimuli with clear emotional distinctions. Despite being a complex black box, where both the subject's mind and deep learning algorithms play a part, differences in data and training can reveal differences among various types of emotional stimuli, aside from simply focusing on the result.

## ABBREVIATIONS

| | |
|---|---|
| **BERT** | Bidirectional Encoder Representations from Transformers |
| **CNN** | Convolutional neural networks |
| **Conv** | Convolutional |
| **DE** | Differential entropy |
| **ECG** | Electrocardiogram |
| **EEG** | Electroencephalogram |
| **EMG** | Electromyogram |
| **E-MFNN** | Emotion-Multimodal Fusion Neural Network |
| **ET** | Eye tracking |
| **FC** | Full connection |
| **GSR** | Galvanic skin response |
| **GAN** | Generative Adversarial Network |
| **ICA** | Independent component analysis |
| **KNN** | K-nearest neighbor |

| | |
|---|---|
| **ReLu** | Linear rectification function |
| **LSTM** | Long short-term memory |
| **RNN** | Recurrent Neural network |
| **RSP** | Respiration |
| **SC** | Skin conductivity |
| **SVM** | Support vector machine |
| **Val** | Validation |
| **VGGNet** | Visual Geometry Group Network |

## ACKNOWLEDGEMENTS

Thanks for the computing support of the State Key Laboratory of Public Big Data, Guizhou University.

### Funding

This study was supported by the National Natural Science Foundation of China (No. 51865003), the Guizhou Science and Technology Plan Project (No. ZK[2021]055, ZK [2023]081, and QKHPTRC[2018]5781), and the Guizhou University Cultivation Project (No. [2019]06). The funders had no role in study design, data collection and analysis, decision to publish, or preparation of the manuscript.

### Grant Disclosures

The following grant information was disclosed by the authors:
National Natural Science Foundation of China: No. 51865003.
Guizhou Science and Technology Plan Project: No. ZK[2021]055, ZK[2023]081, and QKHPTRC[2018]5781.
Guizhou University Cultivation Project: No. [2019]06.

### Competing Interests

The authors declare that they have no competing interests.

### Author Contributions

- Zhuen Guo conceived and designed the experiments, performed the experiments, performed the computation work, authored or reviewed drafts of the article, and approved the final draft.
- Mingqing Yang conceived and designed the experiments, performed the experiments, prepared figures and/or tables, authored or reviewed drafts of the article, and approved the final draft.
- Li Lin conceived and designed the experiments, performed the experiments, performed the computation work, authored or reviewed drafts of the article, funding acquisition, and approved the final draft.

- Jisong Li performed the experiments, analyzed the data, prepared figures and/or tables, and approved the final draft.
- Shuyue Zhang performed the experiments, performed the computation work, authored or reviewed drafts of the article, and approved the final draft.
- Qianbo He performed the experiments, analyzed the data, authored or reviewed drafts of the article, and approved the final draft.
- Jiaqi Gao performed the experiments, performed the computation work, authored or reviewed drafts of the article, and approved the final draft.
- Heling Meng performed the experiments, performed the computation work, authored or reviewed drafts of the article, and approved the final draft.
- Xinran Chen performed the experiments, analyzed the data, prepared figures and/or tables, and approved the final draft.
- Yuehao Tao performed the experiments, prepared figures and/or tables, and approved the final draft.
- Chen Yang performed the experiments, prepared figures and/or tables, and approved the final draft.

## Data Availability
The data and code are available at figshare: Guo, Zhuen (2023). E-MFNN. figshare. Dataset. https://doi.org/10.6084/m9.figshare.24866613.v1.

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
