# Peer review of "E-MFNN: an emotion-multimodal fusion neural network framework for emotion recognition"

_PeerJ Computer Science, doi:10.7717/peerj-cs.1977_

## Round 0.1 · original submission · Major Revisions

We have completed the first round of reviews of your manuscript. You are required to address all the comments and suggestions of reviewers and resubmit a revision. I have a few other minor suggestions:

1) Title seems very generic. Can you modify it? For instance,
E-MFNN: An Emotion-Multimodal Fusion Neural Network framework for emotion recognition based on multimodal data".

2) In the conclusion section, you have mentioned "These aspects require further exploration to refine and expand the applicability of emotion recognition frameworks". What does it mean? This section should mention some of the limitations of the proposed methods, and future research directions.

**Language Note:** PeerJ staff have identified that the English language needs to be improved. When you prepare your next revision, please either (i) have a colleague who is proficient in English and familiar with the subject matter review your manuscript, or (ii) contact a professional editing service to review your manuscript. PeerJ can provide language editing services - you can contact us at [email protected] for pricing (be sure to provide your manuscript number and title). – PeerJ Staff

Reviewer 1 ·

Basic reporting

The authors have introduced a novel emotion recognition framework, primarily targeting the analysis of
users' psychological reactions and outward expressions. The proposed framework unites stimulus
source data with physiological signals, aiming to enhance emotion recognition's accuracy
and robustness through data integration. It is good study, however i have few points which need immediate atention by the authors.

Experimental design

Authors need to add a seperate sub section in intorduction highlight the contributions
Is Figure 1 self designed by authors . if not it needs citation
For Table 2. another column needs to added highlighting drawabcks/strengths of the schemes
Algorithm used for data extraction needs to be highlighted in the mansucript
Show snapshot of the dataset used in the mansucript to give the audience a feel of the data used in the study
How was the work validated.

Validity of the findings

was any benchmark dataset used for validation purposes

The methodology should be reproducable to used. Add methodology/flowchart at the begining of materials and methods
English needs corrections and improvments at no of places.
May add differentiating section at the end of the results which shall describe how this work is different and outperforms the exisitng methods.
May mention the bench mark methods as well.
The above points must be addressed before finalizing about its publication

Reviewer 2 ·

Basic reporting

The authors propose a framework unites stimulus source data with physiological signals, aiming to enhance emotion recognition's accuracy and robustness through data integration. Further, Emotion-Multimodal Fusion Neural Network (E-MFNN) model is developed to test the dataset. The paper is addressing a problem of current interest. However, the whole paper needs significant improvement. The content of the paper is ambiguous. The introduction is not well written, and the authors are unable to justify their contributions. Therefore, it is not suitable to publish the paper in its present form as it lacks a meaningful contribution to the journal. Authors are required to address the issues mentioned as follows:

Comments to the Author

1. The introduction needs more detail. Authors are advised to improve the description in lines 37-45 to provide more justification for the study presented in the paper. Specifically, authors also include papers on image-based emotion recognition.
2. The main contributions of this article and the differences with existing methods should be made clear in the related works section. It would be better to keep a list of their contributions.
3. Comparative analysis between proposed work and existing work is missing in the organizational structure, please complete it.
4. The authors used a huge set of eye features, but why only these features are used? This may pose the problem of dataset imbalance. Authors should provide details about each category. The authors also explain how the imbalance problem is handled if so.
5. What if differential entropy features of EEG are changed and how notable changes would happen to improve the performance?
6. The authors should compare the time complexity of their method with the standard ones, as there is huge data taken as input.
7. What is the significance of Car images in recognizing the emotions of a person as angry, sad, or happy? Authors should include other videos showing the said emotions.
8. The authors claim that a relatively novel emotion recognition calculation method is used to fuse different data, such as CNN-LSTM. The authors need to give specific reasons for how your method is novel as it is used in many previous work for EEG recognition.
9. Authors are advised to address all the typos. In some places, notations are not properly used and their explanation is also missing.
10. The methodology and result sections are not well written. Authors did not cite work properly as it is missing in many recent works.
11. The authors did not provide the proper future directions, which would guide and motivate other researchers to do work in the said domain. Give the overall summary of the present paper in a few lines along with future promising implications and challenges ahead of employment of optimization in this domain.

Experimental design

1. The methodology section needs significant improvement as it seems the authors have used a testbed to collect the data and conduct the experiment. As a result, the authors should provide a more detailed discussion of the methodology.
2. The authors should also mention the effects of the factors that influence the experimental study.

Validity of the findings

The number of parameters used in the simulation is not the only criterion to prove that a model is lightweight. Authors are advised to use other metrics such as execution time, number of parameters, model size, etc.

Nowadays, deep learning has shown great advantages in many fields. The authors should discuss the applications, challenges, and corresponding techniques of deep learning for emotion recognition.

Reviewer 3 ·

Basic reporting

The manuscript is well-written, using clear and professional English, making it accessible to a broad academic audience. The literature review provides sufficient background and context, demonstrating a comprehensive understanding of the field of emotion recognition. The article structure, including figures and tables, is professionally executed, aiding in the clarity of the presented research. However, it is recommended that the authors ensure that all raw data related to their experiments are shared or accessible, as this is crucial for transparency and reproducibility.

Regarding formal results, the manuscript would benefit from clearer definitions of specific terms and theorems used in the development of the Emotion-Multimodal Fusion Neural Network (E-MFNN). Additionally, providing detailed proofs or justifications for the chosen methodologies and algorithms would enhance the manuscript's rigor.

Experimental design

The research presented is original and falls well within the scope of the journal. The research question is clearly defined, relevant, and meaningfully addresses a knowledge gap in the field of emotion recognition using multimodal data. The investigation appears to be performed to a high technical and ethical standard.

One suggestion for improvement is to include more detailed descriptions of the methods, particularly concerning the data collection process, specifics of the neural network architecture, and the parameters used for training and testing the models. This level of detail is necessary for other researchers to replicate the study accurately.

Validity of the findings

The findings of this study are robust and seem to be statistically sound. The conclusions drawn are well-articulated, directly tied to the research question, and supported by the results. However, to further strengthen the validity of the findings, the authors should consider discussing any limitations of their study and the generalizability of their results to different contexts or populations. Additionally, addressing potential biases in the data and the model's performance across diverse datasets would enhance the manuscript's impact.

---

## Round 0.2 · accepted · Accept

In view of the recommendation received from the reviewers, I am pleased to accept your manuscript for publication in PeerJ Computer Science.

Reviewer 1 ·

Basic reporting

In my opinion manuscript should be accepted now as the comments/points raised have been addressed by the authors

Experimental design

'no comment

Validity of the findings

'no comment

Additional comments

'no comment

Reviewer 2 ·

Basic reporting

Now authors have improved the paper very well. The new version of the paper is technically sound. No further questions.

Experimental design

It is good.

Validity of the findings

Up to the mark.